# Oxygen-Enhanced R2* Weighted MRI and Diffusion Weighted MRI of Head and Neck Squamous Cell Cancer Lymph Nodes in Prediction of 2-Year Outcome Following Chemoradiotherapy

**DOI:** 10.3390/cancers17142333

**Published:** 2025-07-14

**Authors:** Harbir Singh Sidhu, David Price, Tim Beale, Simon Morley, Sola Adeleke, Marianthi-Vasiliki Papoutsaki, Martin Forster, Dawn Carnell, Ruheena Mendes, Stuart Andrew Taylor, Shonit Punwani

**Affiliations:** 1Centre for Medical Imaging, University College London, 2nd Floor Charles Bell House, 43-45 Foley Street, London W1W 7TS, UK; olusola.adeleke@kcl.ac.uk (S.A.); v.papoutsaki@ucl.ac.uk (M.-V.P.); stuart.taylor1@nhs.net (S.A.T.); 2Imaging Department, University College London Hospitals, 235 Euston Road, Ground Floor North, London NW1 2BU, UK; tim.beale@nhs.net (T.B.); simon.morley3@nhs.net (S.M.); 3Medical Physics and Biomedical Engineering, University College London Hospitals, 235 Euston Road, Basement, London NW1 2BU, UK; 4Department of Oncology, University College London, UCL Cancer Institute, Paul O’Gorman Building, 72 Huntley Street, London WC1E 6DD, UK; m.forster@ucl.ac.uk; 5Radiotherapy Department, University College London Hospitals, 235 Euston Road, Basement Floor, London NW1 2BU, UK; dawn.carnell@nhs.net (D.C.); r.mendes@nhs.net (R.M.)

**Keywords:** head and neck cancer, MRI, translation, functional imaging

## Abstract

Head and neck squamous cell cancer (HNSCC) with involved lymph nodes (LNs) are often treated with chemoradiotherapy (CRT). Historically, invasive direct measurements showed that LNs can be hypoxic (low in oxygenated blood), and this may be associated with an increased risk of recurrence after CRT. We measured the oxygenation of these LNs non-invasively using MRI (using T2* MR sequences) to test this relationship. We found that, contrary to prior direct measurements, our results suggested that ‘hypoxic’ LNs at baseline tended not to develop recurrence, and furthermore, LNs that became more hypoxic on 100% oxygen were also less likely to develop recurrence.

## 1. Introduction

Head and neck squamous cell carcinoma (HNSCC) is the seventh most common cancer globally, accounting for more than 660,000 new cases, with approximately 60% of patients presenting with locally advanced non-metastatic disease, and is responsible for 325,000 deaths annually [1]. The presence of cervical lymph node metastases is an important adverse prognostic factor [2]. Over the past decade, organ preservation strategies employing CRT have become an accepted alternative to surgery [3], with 5-year overall survival rates of 30 to 50% [4], though associated with significant toxicity [5]. Locoregional failure is the predominant pattern of post-CRT relapse occurring in 25–30% [6], and the pre-treatment identification of resistant tumours could facilitate treatment modification and/or tailored monitoring. Current prognostication strategies include accurate staging, clinical features [7], and tumour biology markers (e.g., preceding oncogenic human papillomavirus infection [8]). MRI is established during diagnosis, staging, and radiotherapy planning, though the ability to discriminate involved lymph nodes remains an area for development [9]. A short axis diameter and the use of morphological criteria (e.g., border regularity) add value in the detection of nodal metastatic disease [10] and has not been applied as a prognostic tool.

Tumour hypoxia, defined as a mismatch between cellular oxygen demand and supply, triggers a cellular response in individual cells after only several hours [11] and often occurs when the distance from a cell to the nearest vasculature is too large for adequate cellular oxygenation [12]. Successful radiotherapy requires oxygen for free radical formation-induced DNA cell death [13]. The invasive direct measurement of tumour oxygenation using an invasive Eppendorf computerised histographic oxygen electrode system [14] has confirmed that HNSCC tumours are hypoxic relative to normal tissue [15] and that increasing tumour hypoxia is associated with poorer post-CRT outcomes [16]. However, an invasive oxygenation assessment is not practical and potentially suffers for sampling error as the needles sample only the very local environment at its tip and ignore the heterogeneity that exists across a tumour, which may account for the observation that directly measured tumour oxygenation is not necessarily an independent predictor of treatment outcome [17].

Nonetheless, other studies have suggested marked tumoral heterogeneity in oxygen distribution, with directly measured hypoxia not necessarily being an independent predictor of treatment outcomes [17]. The non-invasive MRI measurement of transverse relaxation time (T2*) has been proposed as a potential marker of tissue oxygenation status [18,19], offering measurements of both hypoxic extent and spatial distribution [20,21,22]. Elevated levels of deoxyhaemoglobin shorten T2*, thereby generating image contrast and giving an indication of tissue hypoxia [23] (though quantitative application has yet to be established [24]), with the potential to detect clinically relevant changes in tumour oxygenation reported in HNSCC [25]. The inverse of the transverse relaxation time, the transverse relaxation rate (1/T2* = R2*), is often used as a more convenient marker.

However, R2* is also affected by static magnetic field gradients that are found at tissue–tissue and tissue–air boundaries. The problem is compounded in the head and neck region due to the complex shape and anatomy to be imaged. Hence, R2* differences between measurements performed on air and when breathing hyperoxic gas could theoretically reflect differences in oxygenation status whilst ameliorating confounding static susceptibility gradient effects. The magnitude of the difference has previously been shown to reflect the tumour hypoxic fraction, as determined by histologic pimonidazole labelling [26], though it is influenced by tumour biology/type [27].

Diffusion weighted imaging (DWI) characterises tissue based on the movement of water within a volume, which depends on microstructural features, such as cellular density, and may be quantified by the apparent diffusion coefficient (ADC). Adverse biological tumour characteristics may decrease the diffusion restriction of water molecules in HNSCC and lead to higher pretreatment ADC. Thus far, however, evidence for the use of pretreatment tumoral ADC in the prediction of sustained response to therapy in HNSCC is mixed [28].

Within this study, we evaluate baseline lymph node (LN) T2* relaxation times; hence, derive the transverse relaxation rate R2* (1/T2*) when breathing air and 100% oxygen to predict chemoradiotherapeutic locoregional response at 2 years in head and neck squamous cell cancer (HNSCC); and compare it against nodal ADC measurements, short axis diameter, and established qualitative radiological features [10].

## 2. Materials and Methods

### 2.1. Study Design

Institutional review board approval was granted, and patients’ informed consent (written and verbal) was obtained prior to entry into this prospective study (R&D No: 09/0327).

Patients aged over 17 years with histologically confirmed HNSCC stages N2/N3 (AJCC TNM Cancer Staging Manual 7th edition; [29]) and due for primary chemoradiotherapy between 10 February 2010 and 31 July 2017, inclusive, were eligible for inclusion (n = 105). Patients who declined trial entry, received incomplete CRT, were unable to undergo MRI, were pregnant, or had prior malignancy were excluded (n = 33). Furthermore, those with incomplete two-year clinical, imaging, and histopathological follow-up data were excluded for subsequent analysis (n = 14), as were those with inadequate/incomplete baseline MR datasets (n = 4). In total, data for 54 patients were accrued (mean age 57.1 years; range 25–79 years) of whom forty-three were male (mean age 57.7 years; range 33–79 years) and eleven were female (mean age 54.8 years; range 25–74 years). The tumour site and stage are summarised in Table 1.

Each patient underwent multiparametric MRI prior to receiving CRT according to the departmental protocol involving intensity modulated radiotherapy (IMRT) with a minimal dose of 60–70 Gy with concurrent cisplatin (or cetuximab where platin contra-indicated) over 6–7 weeks. Patients with tonsillar SCC underwent tonsillectomy prior to treatment. Patients were followed-up with to determine sustained local complete response to therapy for two years or otherwise, and responders and non-responders were compared as noted below.

### 2.2. Multiparametric Magnetic Resonance Imaging

Subjects underwent multiparametric MRI using a 1.5T static magnet (Avanto, Siemens, Erlangen, Germany) with carotid coils in supine position with median interval to CRT of 22 days. Anatomic axial T2 weighted sequences covered the base of the skull to upper thorax. Diffusion weighted imaging was then performed using six b-values (0, 50, 100, 300, 600, and 1000 s/mm^2^). Axial T2* weighted imaging was undertaken using a multiple gradient echo sequence (echo times 12, 24, 36, and 48 ms). T2* images were initially acquired with the patient breathing air and then repeated after 100% oxygen inhalation at 15 L/min for four minutes via a non-rebreather facemask prior to scanning, which continued for the duration of these sequences [30]. The full MRI parameters are given in Table 2.

### 2.3. Image Analysis

Multiparametric MR images were evaluated by two experienced head and neck radiologists (XX and YY, with 16 and 9 years of experience, respectively; blinded for review) in consensus to identify pathologically involved head and neck nodes with reference to all prior imaging and cytology/histology, though blinded to 2-year outcome. The short axis diameters of each involved node; the largest short axis diameter node for each patient; and qualitative morphological parameters for each lymph node as binary descriptors of nodal contour (ovoid/round), margins (smooth/irregular), enhancement pattern (diffuse/heterogenous), and necrosis (present/absent) were recorded. T2* maps for both ‘air’ and ‘100% oxygen’ were produced from the multiple gradient echo sequences by the previously described numerical fitting algorithm [31] using Matlab (version 7.13, MathWorks Inc., Natick, MA, USA). ADC maps were produced using monoexponential least-squares fitting incorporating all b-values.

A third radiologist (ZZ, with 7 years of experience; blinded for review) aware of the location of each pathological node though unaware of the parametric maps or follow-up data referenced the anatomical images to volumetrically contour the pre-defined nodes on both ‘air’ and ‘100% oxygen’ 12 ms echo T2* weighted images using dedicated software (Jim 5.0, Xinapse systems, Thorpe, Waterville, UK) excluding areas of necrosis. These segmented volumes were then transferred to the air and 100% oxygen T2* parametric maps. Secondly, the radiologist volumetrically contoured the identified nodes on b300 diffusion images, and these segmented volumes were transferred to the monoexponential ADC parametric maps for each patient (Figure 1).

In total, 170 nodal volumes of interest (median 3/patient; range 1–13 nodes) were contoured for each of the ‘air’ and ‘100% oxygen’ T2* maps and the ADC maps. The median, skewness, and kurtosis were derived for each nodal volume from both the T2* and ADC maps.

### 2.4. Treatment Outcome Categorisation

A multidisciplinary review of at least two years of clinical, radiological, and histopathological follow-up data was performed by at least one of each of the following clinicians: an experienced head and neck radiologist (XX and/or YY), a treating clinical oncologist (AA and/or BB), a histopathologist with expertise in head and neck oncology, and an ENT surgeon. Categorisation into two groups was achieved by consensus: 59% of patients demonstrated sustained post-CRT complete local response (CR; n = 32/54; total 104/170 nodes) with a mean age of 60.5 years (range 39 to 79 years), whilst 41% of patients developed local nodal disease relapse (RD; n = 22/56; total 66/170 nodes), with a mean age of 57.2 years (range 25 to 75 years).

### 2.5. Statistical Analyses

All analyses compared the two outcome groups (i.e., CR versus RD) to assess for significant pretreatment differences between short axis diameter, qualitative descriptors, and histographic R2* and ADC metrics and were performed on both a largest node per patient (LNPP) and an all-node (AN) basis.

An LNPP analysis compared short axis diameters using the two-tailed Mann–Whitney U test and qualitative classifiers using Fisher’s exact test. LNPP pairwise parameter differences between the air and 100% oxygen R2* datasets were calculated using the Wilcoxon signed rank test. The histographic LNPP R2* and ADC parameter differences were compared between outcome groups using the Mann–Whitney U test. Statistical significance was assigned at *p* values below 0.05 performed using SPSS statistics for Windows (version 16; IBM, Armonk, NY, USA). A univariate analysis was also performed. Due to the binary outcome measures, all these analyses were performed using logistic regression. Subsequently, the joint association between variables and the outcome was examined in the multivariable logistic regression analysis. To restrict the number of variables in this stage of the analysis, only factors with univariable *p*-values of <0.1 were included.

For the AN analysis, baseline morphological characteristic differences were assessed using Fisher’s exact test. Nodal short axis diameter and histographic R2* and ADC differences were analysed using a linear mixed model (Stata v13, StataCorp LLC, College Station, TX, USA) to account for multiple samples per patient. Fitting was performed using response as the fixed factor and patient as the random factor. Means and 95% confidence intervals (CIs) were estimated from the model for each patient response group. Where the data was right-skewed, the log measurement was used. Where the data was right-skewed, including negative values, a small constant was added to all values before the log transformation was applied, with estimated means and 95% CIs backtransformed.

For the analysis of paired air and 100% oxygen measurements within patient groups, a linear mixed model was fitted using measurement type (air or 100% oxygen) as the fixed factor, with patient and node as the random factors. Means and 95% confidence intervals were estimated from the model for each measurement type in the same manner.

## 3. Results

### 3.1. Patient Cohort

In total, the data for 54 patients were accrued (mean age 57.1 years; range 25–79 years), of whom forty-three were male (mean age 57.7 years; range 33–79 years) and eleven were female (mean age 54.8 years; range 25–74 years). Each patient underwent 1.5T MRI (per the Section 2) prior to receiving CRT according to the departmental protocol involving intensity modulated radiotherapy (IMRT) with a minimal dose of 60–70 Gy with concurrent cisplatin (or cetuximab where platin contra-indicated) over 6–7 weeks.

Patients were followed-up for at least two years and, by multidisciplinary consensus (see Section 2), were categorised into two groups: 59% of patients demonstrated sustained post-CRT complete local response (CR; n = 32/54; total 104/170 nodes), with a mean age of 60.5 years (range 33 to 79 years), whilst 41% of patients developed local nodal disease relapse (RD; n = 22/56; total 66/170 nodes), with a mean age of 57.2 years (range 25 to 75 years). Individual pathological nodes were volumetrically contoured to parametric T2* and ADC maps, as noted below, and comparisons were made between the two groups of patients both on the ‘largest node per patient (LNPP)’ and ‘all-node (AN; 170 LN total)’ bases.

### 3.2. Largest Node per Patient Analysis

The results of this analysis are summarised in Table 3.

#### 3.2.1. Short Axis Diameter and Qualitative Descriptors

The mean short axis diameter was slightly larger for the RD group, though this difference was not significant (19 mm for the CR group and 23 mm for the RD group; *p* = 0.21). There were no significant differences in the binary qualitative radiological descriptors between the CR and RD groups (*p* values between 0.17 and 0.55). For the univariate analysis, the size of the association between each factor and a response is quantified by the odds ratios, presented with corresponding confidence intervals. For the categorical variables, these represent the odds of a response in each category relative to the odds in a baseline category. No strong predictors of maintained response were identified here.

#### 3.2.2. R2* Parameters

A pairwise comparison revealed a significant increase in median R2* values when breathing 100% oxygen compared to breathing air in the CR group (mean R2*air = 25.6 ms, R2*O_2_ = 27.6 ms; *p* = 0.012); in the RD group, whilst there was also a trend in lengthening the median T2*, this was not significant (*p* = 0.055) and the changes are illustrated in a line graph for each largest node per patient in Figure 2. There were no significant pairwise differences for histogram R2* skewness or kurtosis (*p* values between 0.33 and 0.60).

When comparing absolute baseline median R2* values between the two groups, the CR group’s largest nodes were significantly longer in R2* times compared to those for the RD group when breathing air (*p* = 0.04) and close to significant when breathing 100% oxygen (*p* = 0.05), illustrated in the box plot in Figure 3. No significant differences were observed in the R2* histogram skewness or kurtosis on air or 100% oxygen between the two cohorts (*p* values between 0.32 and 0.64).

For the univariate analysis, the odds ratio of these continuous variables indicate the relative change in the odds of a response for every one-, five- or ten-unit increase in that parameter (dependent on the size of the difference). These showed a similar relationship of these parameters to outcome, albeit less marked, with the mean R2* on air being significant (and O_2_ approaching significance).

#### 3.2.3. Diffusion Weighted Imaging Parameters

There were no significant differences between the largest nodal pretreatment median ADC values between the CR (ADC = 0.91 × 10^−3^ mm^2^/s) and RD (ADC = 0.89 × 10^−3^ mm^2^/s) groups (*p* = 0.99). Furthermore, histogram skewness and kurtosis were also not significantly different between the two groups (*p* = 0.63 and 0.72, respectively).

The univariate analysis on these continuous variables also demonstrated no strong associations with ultimate response, per the odds ratios.

A multivariate analysis of the LNPP data was also undertaken, and the data suggested very highly correlations, and thus collinearity, between all the main air parameters that were associated with a response. The regression model, however, runs into problems with all these variables, taking the mean R2* on air and oxygen and necrosis as the best-performing predictors in this model. A backward selection procedure was used to retain only the most important variables in the final model. However, whilst there was some evidence that mean R2* on air was associated with sustained response, this did not reach statistical significance (OR 1.33, 0.96–1.86; *p*-value of 0.07).

### 3.3. All-Node Analysis

The results for this linear mixed model analysis are summarised in Table 4.

#### 3.3.1. Short Axis Diameter and Qualitative Descriptors

The mean short axis diameter of all pathological nodes was slightly larger for the RD group, though this difference was not significant (12.0 mm for 104 nodes in the CR group and 13.5 mm for 66 nodes in the RD group; *p* = 0.10). There were no significant differences in the binary qualitative radiological descriptors between the CR and RD groups (*p* values between 0.11 and 0.64).

#### 3.3.2. R2* Parameters

The all-node pairwise comparison again revealed a significant increase in median R2* values when breathing 100% oxygen compared to breathing air in the CR group (mean R2*air = 26.4 s^−1^, R2*O_2_ = 28.1 s^−1^; *p* = 0.0006), and in the RD group, whilst there was also a trend of lengthening the median R2* times; again, this was not significant (*p* = 0.14). There were no significant pairwise differences for histogram R2* skewness or kurtosis (*p* values between 0.05 and >0.99).

When comparing absolute all-node baseline median R2* values between the two groups, the CR group nodes showed significantly lower R2* values compared to the RD group when breathing air (*p* = 0.049), though this difference was not significant on 100% oxygen (*p* = 0.07). No significant differences were observed in all-node R2* skewness or kurtosis on air or 100%-oxygen between the two cohorts (*p* values between 0.15 and 0.91).

#### 3.3.3. Diffusion Weighted Imaging Parameters

The all-node DWI comparison revealed that there were no significant differences (*p* = 0.91) between the pretreatment median ADC values between the CR group (ADC = 0.86 × 10^−3^ mm^2^/s) and the RD group (ADC = 0.88 × 10^−3^ mm^2^/s). Furthermore, ADC histogram skewness and kurtosis were also not significantly different between the two groups (*p* = 0.26 and 0.97, respectively).

## 4. Discussion

This study has demonstrated that cancerous lymph nodes in patients which subsequently respond to CRT demonstrate significant lengthening of R2* relaxation times when switching from breathing air to 100% oxygen, whilst nodes in patients who relapse within two years do not suggest a paradoxical increase in deoxyhaemoglobin on breathing 100% oxygen in nodes which respond favourably to CRT. To our knowledge, this is the first study to examine R2* changes exclusively within HNSCC lymph nodes in response to hyperoxic gas compared with air correlated with CRT response. The second finding is that responding nodes had significantly longer baseline R2* times on air compared with relapsing nodes, which may imply more pretreatment hypoxia in responding nodes. Both of these findings are potentially counterintuitive given prior direct measurement hypoxia studies as well as several preclinical and early clinical studies evaluating oncological R2* imaging.

It is known that whilst the R2* of the vascular space is dependent on fractional blood oxygenation, it is also affected by other factors and is a quadratic function of haematocrit levels and magnetic field strength [32]. Whilst direct inference of tissue hypoxia based on quantitative R2* measurements is likely problematic (though a moderate correlation with directly measured oxygen tension has been observed in prior validation studies [21]), differential response to breathing hyperoxic gas circumvents many of these additional confounding factors by evaluating the significance of relative R2* changes in the same sitting and thus will likely be the more important finding. Differences in R2* contrast in cancerous LNs to hyperoxic challenge between prior studies and the current findings may potentially be explained by the vascular properties of these nodes (most studies’ focus or a combination with primary tumours) as well as the use of 100% oxygen without hypercapnia.

The magnitude of R2* changes within tumours on breathing hyperoxic gas have been shown to mirror tumour hypoxic fraction changes, as determined by pimonidazole labelling [26], though in preclinical studies, the changes were not directly proportional to the absolute measured tissue oxygenation [33]. Kotas et al. [23] examined the changes in T2* values (1/R2*) in HNSCC tumours in response to breathing 2% CO2/98% oxygen gas and 100% oxygen and demonstrated no significant tumoral T2* differences when breathing either the hyperoxic gas mixture or air. However, this study was on a relatively small cohort (13 patients) and used a combined assessment of primary HNSCC tumours with involved lymph nodes (only six lymph nodes were examined). Other studies have examined the effect of breathing hyperoxic hypercapnic gas (2% CO2/98% oxygen) on T2* times in HNSCC [34], though they examined primary tumours rather than lymph nodes, and demonstrated lengthening in T2* times (shortening of R2*) with this gas mixture in eleven primary HNSCC tumours, inferring increasing oxygenation on hyperoxic gas.

Unlike these hypercapnic challenges, 100% oxygen, as used in our study, will cause different physiological effects on respiration, oxygenation, and blood flow. It has been noted that breathing hyperoxic gas results in heterogenous effects on tumour blood flow [35], which may result in a ‘steal’ effect, whereby blood flow may be increased at the expense of an adjacent location [36]. T2* measurement using multiple gradient echoes should be relatively robust to blood flow variation [37]; however, perturbation due to vasomodulation can alter the effective concentration of blood in tissues [35], and a decrease in R2* rate will therefore only occur in hypoxic tumours with functional vasculature that respond in a certain manner to hyperoxic challenge [38]. A preclinical study by McPhail and Robinson [39] correlated R2* measurement in a breast tumour murine model validated with pimonidazole and other quantitative histological markers of hypoxia.

This demonstrated that tumours with a faster baseline R2* and a larger decrease in R2* with carbogen breathing were more vascularised and had a greater functional blood volume than tumours with a slower baseline R2* and negligible response to carbogen; functional vasculature was central to the observed R2* contrast changes and was said to be dominated by blood volume. A clinical study has further reported a significant inverse correlation between breast tumour R2* and grade [40]. It was also noted that both benign and necrotic tumours were omitted from this clinical study, as necrosis can cause a paradoxical decrease in R2* owing to the lack of erythrocyte delivery [41].

A recent human prostate cancer study [42] has demonstrated that tumours exhibiting low R2* times stained positive, whilst those with high R2* were negative to pimonidazole staining, which was attributed to tumour regions occurring in tissues which were highly vascularised. It is increasingly clear that, in the context of hypoxia, R2* measurement can vary widely and may be highly dependent on the tumour examined and its vasculature, and further evaluation of the tissue bases of these contrast mechanisms is required. Nonetheless, this first demonstration of a significant difference in R2* response to hyperoxic gas (and potentially in the baseline R2* characteristics) of responding nodes compared with those which relapse within two years of CRT is in itself important. We posit that the underlying observed R2* differences between the outcome groups may thus relate to lymph node-specific blood flow/vascular mechanisms as well as hypoxia, which would potentially be useful for exploitation in biomarker development.

The requirement for predictive HNSCC biomarker development is reinforced by the demonstration that pre-treatment nodal short axis diameter and various conventionally employed qualitative morphological descriptors of malignancy were, perhaps unsurprisingly, not useful as discriminators of subsequent successful response to CRT.

Furthermore, this study also demonstrated that baseline diffusion weighted imaging parameters derived from monoexponential ADC were not significantly different between the nodes demonstrating sustained local complete post CRT response and those that subsequently relapse. Again, this is perhaps not unexpected as there have been conflicting results in other studies. A previous study specifically examining nodal HNSCC failure after CRT demonstrated significantly higher mean pre-treatment ADCs than in relapsing nodes compared with those that were controlled [43].

Other studies have found that, though there were trends towards higher pretreatment ADC predicting treatment failure, the relationship was not significant [44], whilst another study demonstrated the opposite relationship based on two-year disease-free survival [45]. Additionally, it should be highlighted that there is a lack of generalizability of DWI data given variable b value selection and, indeed, variability of the ADC values across MRI systems and sequences [46].

This study benefits from the prospective design and length of follow-up; however, it also has some limitations that should be considered. There was an inherent bias potentially derived from the different cancer locations, stages, and treatments received by the patients included in this study. The sample size was relatively small, though the analysis of multiple nodes in both groups per patient (with linear mixed modelling to account for sampling) did recapitulate the largest node per patient differences between the CR and RD groups. Independent verification of nodal oxygenation was not undertaken (e.g., using invasive polarographic electrode measurement or histological verification), though it would probably not be feasible in this cohort given the number of nodes examined. It was also difficult to account for true physiological oxygenation fluctuations in the blood flow within the tumour capillary network that can lead to well-described transient or cyclical hypoxia [47]. As noted previously, there is considerable heterogeneity within tumour volumes, particularly with respect to R2*, and under hyperoxic breathing [23]—the use of volumes (rather than single axial regions) of interest was employed to mitigate this effect as far as possible. Previous clinical R2* studies have employed between 4 and 16 echo times, though it has been demonstrated that the signal acquired with GRE sequences can be dominated by noise for longer echo times [25].

## 5. Conclusions

This study demonstrates that the significant differential response to 100% oxygen and higher baseline R2* measures between responding versus non-responding HNSCC lymph nodes could be exploited in risk stratification prior to CRT given that traditional qualitative radiological classifiers of nodal pathology and DWI parameters at baseline were poor discriminators of subsequent local response. Future work is, however, required to understand the contrast mechanisms of R2* imaging underpinning these observed differences in the context of hypoxia and would benefit from a more formal evaluation of confounding factors (e.g., blood volume fraction and macroscopic field homogeneity) as well as direct histological verification in the HNSCC setting.

## Figures and Tables

**Figure 1 cancers-17-02333-f001:**
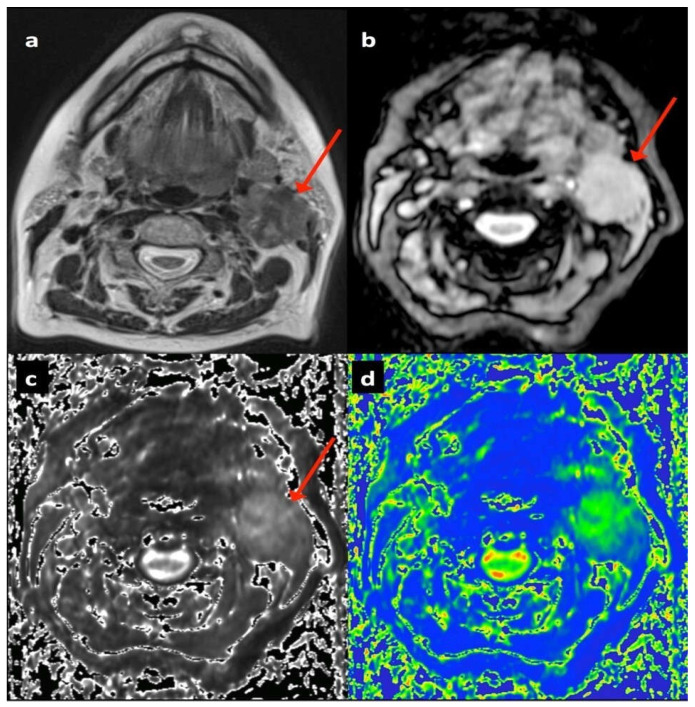
Seventy-four-year-old female patient with T2 N2b left tongue base squamous cell carcinoma showing left upper deep cervical nodal mass (arrowed): (**a**) T2-TSE axial, (**b**) T2* gradient echo (12 ms) on air, (**c**) T2* parametric map on air, (**d**) diffusion weighted b300 image.

**Figure 2 cancers-17-02333-f002:**
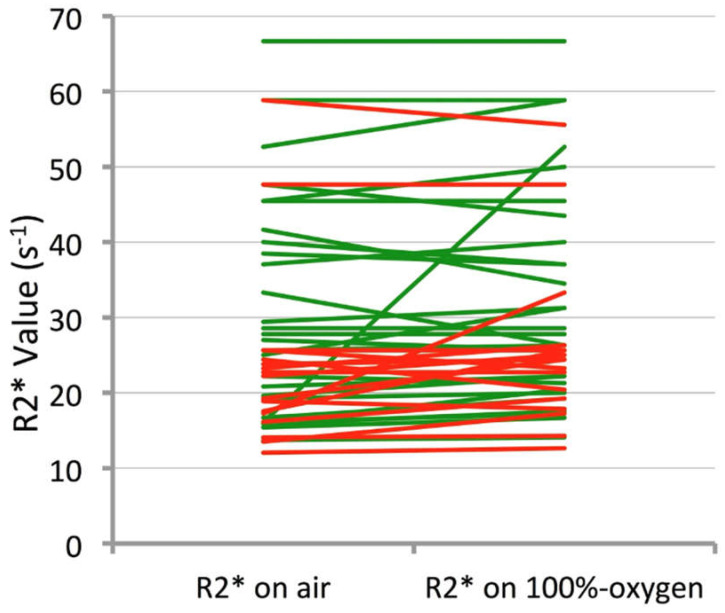
Line chart demonstrating change in R2* from air to 100% oxygen in the largest node per patient for each CR patient (represented by a single green line) and RD patient (represented by the red lines).

**Figure 3 cancers-17-02333-f003:**
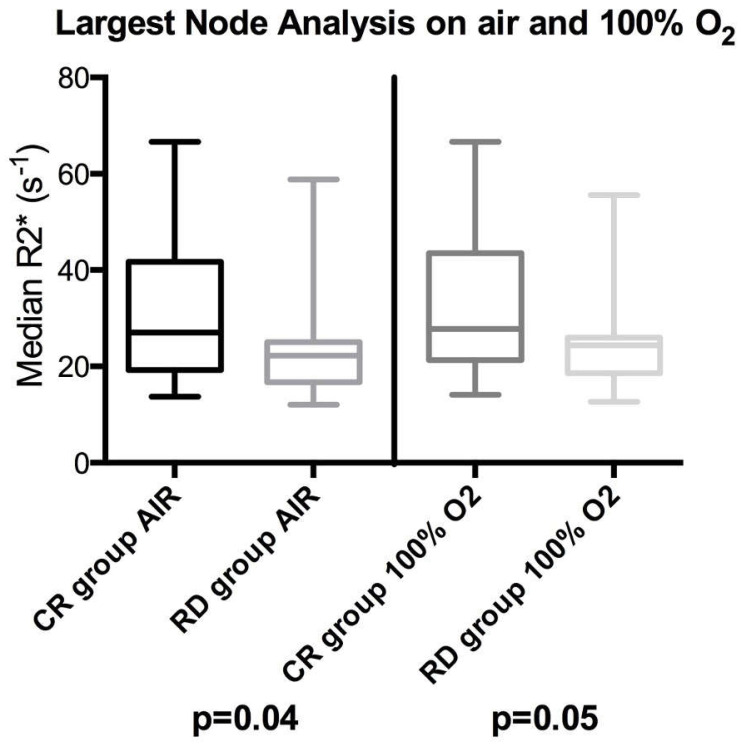
Box plots showing differences between complete responding pathological lymph node (CR) and residual/recurrent lymph node disease relapse (RD) groups’ median R2* distributions on a largest node per patient basis. The left side shows the node analysis on air for the largest node and the right side on 100% oxygen. The box indicates the interquartile range, line median, and whiskers most deviated range. Mann–Whitney U *p*-values are shown beneath each plot.

**Table 1 cancers-17-02333-t001:** Patient demographics, tumour site and stage (N2a: single > 3 cm but ≤6 cm in greatest dimension, N2b: metastases in multiple ipsilateral nodes < 6 cm, N3: node > 6 cm), p16 status (n/a—not measured, + positive, − negative) and treatment received (neo/induction chemotherapy). Green column; sustained response, red; eventual relapse.

Maintained Complete Response at 2 Years				Relapse of Disease Within 2 Years			
Age/ Gender	Primary Site and Stage	p16	Treatment	Age/ Gender	Primary Site Stage	p16	Treatment
67 M	Hypopharynx T3N2bM0	n/a	Neo: cisplatin/5FU Chemo: cisplatin Radiation: IMRT	70 M	Oropharynx T4N2M0	n/a	Neo: nil Chemo: cetuximab Radiation: IMRT
55 F	Soft palate T2N2cM0	n/a	Neo: cisplatin/5FU Chemo: cetuximab Radiation: IMRT	60 M	Pyriform fossa T3N2bM0	n/a	Neo: cisplatin/5FU Chemo: cetuximab Radiation: IMRT
74 F	Tonsillar (fauc) T1N3M0	+	Neo: cisplatin/5FU Chemo: cisplatin Radiation: IMRT	64 F	Soft palate T1N2bM0	−	Neo: cisplatin/5FU Chemo: cetuximab Radiation: IMRT
56 M	Tonsil T1N2M0	n/a	Neo: cisplatin/5FU Chemo: cisplatin Radiation: IMRT	48 M	Pyriform fossa T2N2bM0	n/a	Neo: cisplatin/5FU Chemo: cisplatin Radiation: IMRT
56 M	Tonsil (fauc) T2N2cM0	+	Neo: nil Chemo: cisplatin Radiation: IMRT	57 M	Tonsil (fauc) T2N2bM0	−	Neo: cisplatin/5FU Chemo: cetuximab Radiation: IMRT
67 M	Tonsil T1N2bM0	+	Neo: cisplatin/5FU Chemo: cisplatin Radiation: IMRT	56 M	Tonsil T3N2bM0	+	Neo: cisplatin/5FU Chemo: cisplatin Radiation: IMRT
56 M	Tongue base T4N2cM0	+	Neo: cisplatin/5FU Chemo: cisplatin Radiation: IMRT	43 M	Tonsil T3N2bM0	+	Neo: cisplatin/5FU Chemo: cisplatin Radiation: IMRT
63 M	Tonsil (fauc) T2N2bM0	+	Neo: cisplatin/5FU Chemo: cisplatin Radiation: IMRT	52 M	Glossotonsillar sulcus T4N2cM0	n/a	Neo: cisplatin/5FU Chemo: cetuximab Radiation: IMRT
52 M	Unknown TxN2M0	+	Neo: cisplatin/5FU (single) Chemo: cisplatin Radiation: IMRT	54 M	Tongue base T3N2cM0	n/a	Neo: cisplatin/5FU Chemo: cisplatin Radiation: IMRT
57 M	Tonsil T2N2bM0	+	Neo: cisplatin/5FU Chemo: cisplatin Radiation: IMRT	64 F	Oropharynx T4N2cM0	n/a	Neo: cisplatin/5FU Chemo: cisplatin Radiation: IMRT
49 M	Tonsil T4N2bM0	+	Neo: cisplatin/5FU Chemo: cisplatin Radiation: IMRT	74 F	Tongue base T2N2bM0	n/a	Neo: nil Chemo: cisplatin Radiation: IMRT
33 M	Tongue base (presumed) TxN2cM0	+	Neo: nil Chemo: cisplatin Radiation: IMRT	49 M	Tongue base T3N2bM0	n/a	Neo: nil Chemo: cisplatin Radiation: IMRT
60 M	Tonsil T4N2bM0	n/a	Neo: cisplatin/5FU Chemo: cisplatin Radiation: IMRT	59 M	Epiglottis T2N2cM0	n/a	Neo: cisplatin/5FU Chemo: cisplatin Radiation: IMRT
47 F	Tonsil T2N2bM0	+	Neo: nil Chemo: cisplatin Radiation: IMRT	44 M	Tonsil T1N2bM0	+	Neo: nil Chemo: cisplatin Radiation: IMRT
46 M	Tonsil T3N2bM0	−	Neo: nil Chemo: cisplatin Radiation: IMRT	63 M	Tongue base T4N3M0	+	Neo: cisplatin/5FU Chemo: cisplatin Radiation: IMRT
62 M	Tonsil T2N2bM0	+	Neo: nil Chemo: cisplatin Radiation: IMRT	49 F	Pyriform fossa T2N2bM0	n/a	Neo: nil Chemo: cisplatin Radiation: IMRT
63 M	Pyriform fossa T3N2bM0	n/a	Neo: nil Chemo: cisplatin Radiation: IMRT	25 F	Tonsil T2N2bM0	n/a	Neo: nil Chemo: cisplatin Radiation: IMRT
67 M	Tonsil T4N2cM0	n/a	Neo: cisplatin/5FU Chemo: cisplatin Radiation: IMRT	75 M	Tongue base T4aN2bM0	−	Neo: cisplatin/5FU Chemo: cisplatin Radiation: IMRT
62 M	Tongue base T4N2cM0	+	Neo: cisplatin/5FU Chemo: cisplatin Radiation: IMRT	64 F	Supraglottic T3N2cM0	n/a	Neo: cisplatin/5FU Chemo: cisplatin Radiation: IMRT
75 F	Tonsil T3N2bM0	n/a	Neo: cisplatin/5FU Chemo: cisplatin Radiation: IMRT	52 M	Pyriform fossa T3N2bM0	n/a	Neo: nil (ALD) Chemo: cisplatin Radiation: IMRT
69 M	Pyriform fossa T2N2cM0	−	Neo: cisplatin/5FU Chemo: cisplatin Radiation: IMRT	68 M	Hypopharnyx T4N2cM0	n/a	Neo: carbopl/5FU Chemo: cetuximab Radiation: IMRT
73 M	Tonsil T3N2bM0	+	Neo: carboplat/5FU Chemo: carboplatin Radiation: IMRT	71 M	Tonsil T4N2bM0	−	Neo: nil Chemo: cisplatin Radiation: IMRT
54 M	Nasopharynx T4N1M0	−	Neo: carboplat/5FU Chemo: carboplatin Radiation: IMRT				
76 M	Hypopharynx T3N2cM0	−	Neo: nil Chemo: nil Radiation: IMRT				
67 M	Tonsil T3N2cM0	+	Neo: carboplat/5FU Chemo: carboplatin Radiation: IMRT				
79 M	Tonsil T1N2bM0	+	Neo: cisplatin/5FU Chemo: cetuximab Radiation: IMRT				
62 M	Tongue base T2 N2c M0	−	Neo: cisplatin/5FU Chemo: cisplatin Radiation: IMRT				
69 M	Tonsil T3N2bM0	+	Neo: cisplatin/5FU Chemo: cisplatin Radiation: IMRT				
51 F	Unknown TxN2bM	−	Neo: cisplatin/5FU Chemo: cisplatin Radiation: IMRT				
72 F	Tonsil T2N2aM0	+	Neo: cisplatin/5FU Chemo: cisplatin Radiation: IMRT				
59 M	Hypopharnyx T4N2bM0	n/a	Neo: cisplatin/5FU Chemo: cetuximab Radiation: IMRT				
41 M	Post-pharynx T4N2cM0	−	Neo: cisplatin/5FU Chemo: cisplatin Radiation: IMRT				

**Table 2 cancers-17-02333-t002:** Multiparametric MRI sequence parameters used for this study. TSE—turbo spin echo, DWI—diffusion weighted imaging, STIR—short tau inversion recovery, GRE—gradient recall echo.

	T2w TSE	T2* GRE	STIR-EPI DWI
**Orientation**	Axial	Axial	Axial
**Repetition time (ms)**	6670	1450	9400
**Echo time (ms)**	93	12, 24, 36, 48	93
**Flip angle (degrees)**	120	25	90
**Bandwidth (Hz/Px)**	250	140	1502
**Field of view (mm)**	180 × 180	195 × 250	206 × 206
**Acquired matrix**	256 × 256	512 × 400	128 × 128
**Slice thickness (mm)**	3	5	4
**Slice gap (mm)**	0.3	2.5	0.4
**Averages**	1	1	4
**Phase encoding direction**	Anteroposterior	Anteroposterior	Anteroposterior
**Fat suppression**	No	No	STIR
**Base matrix**	256	256 × 100	128
**Number of acquisitions**	1	1	1
**b-values (s.mm^−2^)**	n/a	n/a	0, 50, 100, 300, 600, 1000
**Total acquisition time (min)**	2 m 54 s	2 min 53 s	2 m 31 s

**Table 3 cancers-17-02333-t003:** Summarising the comparison between complete responding (CR) and residual/recurrent lymph node disease (RD) groups on a ‘largest node per patient’ (LNPP) basis. The yellow cells indicate differences which are significant for either analysis. For the univariate analysis, (*) odds ratios are given for a 5-unit increase in the variable, and (**) odds ratios are given for a 10-unit increase in the variable.

PARAMETER		TREATMENT OUTCOME GROUP		*p*-Value/Odds Ratio	
		Complete Response (CR)	Residual/Recurrent Nodal Disease (RD)		
**Number of patients (percentage)**		32 (59%)	22 (41%)		
**Qualitative Descriptors**					
				**Fisher’s exact**	**Univariate Odds Ratio (95% CI)**
**Contour**	Round	16 nodes	9 nodes	0.55	1
	Ovoid	16 nodes	13 nodes		1.65 (0.53, 5.18)
**Margins**	Irregular	20 nodes	10 nodes	0.17	1
	Regular	12 nodes	12 nodes		1.40 (0.37, 5.24)
**Enhancement**	Heterogenous	20 nodes	14 nodes	0.39	n/a
	Diffuse	11 nodes	8 nodes		
**Necrosis**	Present	19 nodes	17 nodes	0.32	1
	Absent	13 nodes	5 nodes		0.32 (0.08, 1.21)
**Quantitative Parameters**					
				**Mann–Whitney**	**Univariate Odds Ratio (95% CI)**
**Mean short axis diameter (mm)**		18.5	22.6	0.21	0.83 (0.57, 1.21) *
**Mean of ADC median values (10^−3^ mm^2^/s)**		0.91	0.89	0.99	0.88 (0.60, 1.29) **
**Mean of ADC skewness**		0.59	0.67	0.63	0.78 (0.27, 2.29)
**Mean of ADC kurtosis**		1.04	1.2	0.72	0.77 (0.29, 2.24)
**Mean of R2* median values (s^−1^)**	Air	25.6	20.2	0.04	1.4 (0.98, 2.04) ** *p* = 0.05
	100% O_2_	27.6	22.5	0.05	1.4 (0.94, 2.12) ** *p* = 0.06
	**Wilcoxon *p*-value**	0.012	0.055		
	**Univariate Odds Ratio (95% CI)**	0.74 (0.55, 1.56) *p* = 0.15			
**Mean of R2* skewness**	Air	0.757	0.483	0.39	1.60 (0.69, 3.70)
	100% O_2_	0.582	0.777	0.32	0.79 (0.28, 2.26)
	**Wilcoxon *p*-value**	0.597	0.356		
**Mean of R2* kurtosis**	Air	1.57	0.489	0.64	1.12 (0.82, 1.56)
	100% O_2_	0.667	1.37	0.44	0.89 (0.64, 1.22)
	**Wilcoxon *p*-value**	0.36	0.33		

**Table 4 cancers-17-02333-t004:** Summarising the comparison between complete responding (CR) and residual/recurrent lymph node disease (RD) groups on an ‘all-node’ (AN) basis using linear mixed models. Data in parentheses indicate 95% confidence intervals: ^a^ log(x) measurement used in the linear mixed model, ^b^ log(x + 1) measurement analysed in the linear mixed model, ^c^ log(x + 1.5) measurement analysed in the linear mixed model, ^d^ log(x + 2) measurement analysed in the linear mixed model. The yellow cells indicate differences that were significant.

PARAMETER		TREATMENT OUTCOME GROUP		*p*-Value
		Complete Response (CR; n = 32 pts/59%)	Residual/Recurrent Nodal Disease (RD; n = 22 pts/41%)	
**Number of nodes (percentage)**		104 (61.2%)	66 (38.9%)	
**Qualitative Descriptors**				
				**Fisher’s exact**
**Contour**	Round	50 nodes	23 nodes	0.11
	Ovoid	54 nodes	43 nodes	
**Margins**	Irregular	50 nodes	38 nodes	0.27
	Regular	54 nodes	28 nodes	
**Enhancement**	Heterogenous	52 nodes	35 nodes	0.64
	Diffuse	52 nodes	31 nodes	
**Necrosis**	Present	48 nodes	38 nodes	0.16
	Absent	56 nodes	28 nodes	
**Quantitative Parameters**				
				**Linear Mixed**
**Mean short axis diameter (mm)**		12.0 (11.0; 13.1)	13.5 (12.1; 15.0)	0.21
**Mean of ADC median values ^a^ (10^−3^ mm^2^/s)**		0.86 (0.81; 0.93)	0.88 (0.83; 0.96)	0.91
**Mean of ADC skewness**		0.72 (0.55; 0.93)	0.58 (0.38; 0.79)	0.26
**Mean of ADC kurtosis ^a^**		3.74 (3.29; 4.13)	3.71 (3.25; 4.23)	0.97
**Mean of R2* median values (s^−1^)**	Air	26.4 (23.6; 29.9)	22.1 (19.5; 25.4)	0.049
	100% O_2_	28.1 (25.1; 31.9)	23.6 (20.8; 27.4)	0.07
	**Pairwise *p*-value**	0.0006	0.14	
**Mean of R2* skewness**	Air ^b^	0.5 (0.4; 0.7)	0.4 (0.3; 0.6)	0.15
	100% O_2_ ^b^	0.5 (0.4; 0.7)	0.6 (0.4; 0.7)	0.91
	**pairwise *p*-value**	>0.99	0.05	
**Mean of R2* kurtosis**	Air ^c^	0.2 (−0.1; 0.6)	0.1 (−0.3; 0.5)	0.51
	100% O_2_ ^d^	0.5 (0.2; 1.0)	0.2 (−0.2; 0.7)	0.26
	**Pairwise *p*-value**	0.36	0.68	

## Data Availability

Data is presented in the paper.

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
