# Peer review of "Oxygen-Enhanced R2* Weighted MRI and Diffusion Weighted MRI of Head and Neck Squamous Cell Cancer Lymph Nodes in Prediction of 2-Year Outcome Following Chemoradiotherapy"

_cancers, 2025, doi:10.3390/cancers17142333_

Round 1
Reviewer 1 Report
Comments and Suggestions for Authors
In this paper, the authors showed that, according to R2*, nodes that have experienced a post-therapeutic complete response are significantly more hypoxic compared to recurrent nodes and, paradoxically, demonstrate a significant increase in hypoxia with 100% oxygen.
1. The article does not describe the study group, except for indicating gender and age, and that all patients had Head and neck squamous cell carcinoma. Provide a detailed description taking into account the localization, stage, type of treatment, number of chemotherapy courses, presence/absence of surgery, radiation therapy, etc.
2. The authors need to conduct a univariate and multivariate analysis to show the significance of the identified parameter for assessing the response to treatment. Provide the odds ratio.
3. In Figure 2, it would be more visual to provide the average values ​​for the group with a complete response and with a partial response/absence of response to treatment.
4. Is hypoxia related only to the response to treatment or to any clinicopathological characteristics of the tumor? e.g. stage, location, etc.?
Author Response
1. The article does not describe the study group, except for indicating gender and age, and that all patients had Head and neck squamous cell carcinoma. Provide a detailed description taking into account the localization, stage, type of treatment, number of chemotherapy courses, presence/absence of surgery, radiation therapy, etc.
Response:
Many thanks- we have updated with a Table (1) which provides the site and stage of disease and also updated the text with respect to treatment (summarised lines 110-120).
2. The authors need to conduct a univariate and multivariate analysis to show the significance of the identified parameter for assessing the response to treatment. Provide the odds ratio.
Response:
We have gone over these data with statistical input- the variables would be continuous rather than discrete ordinal outcome variables. This would require us to logistically regress the data or impose some kind of cut-off which would perhaps be less valuable than the significance values presented and more prone to error (and more so with the all nodes analysis with multiple samples/per patient which we have already logistically regressed when performing univariate and multivariate analyses where skewed).
3. In Figure 2, it would be more visual to provide the average values ​​for the group with a complete response and with a partial response/absence of response to treatment.
Response:
Many thanks; this is provided in the Tables (for purpose of visual representation we feel this shows the range of values/changes).
4. Is hypoxia related only to the response to treatment or to any clinicopathological characteristics of the tumor? e.g. stage, location, etc.?
Response:
Hypoxia was related to sustained response vs eventual relapse; there was not relationship to clinicopathological characteristics though the assessment of this was not particularly easy/useful given the various locations and stages of the tumours involved.
Reviewer 2 Report
Comments and Suggestions for Authors
The study by Sidhu et al. provides novel evidence on the use of oxygen-enhanced R2* magnetic resonance imaging as a potential biomarker to predict response to chemoradiotherapy in head and neck cancer.
The study is interesting and well conducted; however, some minor comments should be considered:
- Ensure consistency in terminology, as CRT is mentioned in both the background and introduction sections.
- Would it be possible to include the histological grade of the tumors evaluated and provide a table with clinical variables such as age, sex, affected lymph nodes, histological grade, and administered CRT?
- Could the strengths of the study be highlighted, such as the prospective design, follow-up period, among others?
Author Response
1. Ensure consistency in terminology, as CRT is mentioned in both the background and introduction sections.
Response:
Many thanks for pointing this out, we have amended as seen in line 53.
2. Would it be possible to include the histological grade of the tumors evaluated and provide a table with clinical variables such as age, sex, affected lymph nodes, histological grade, and administered CRT?
Response:
Yes, this feedback was universal. We have included new Table 1 outlining this and amended lines 110-120 to include details of upfront surgery in tonsillar SCC and treatment.
3. Could the strengths of the study be highlighted, such as the prospective design, follow-up period, among others?
Response:
Many thanks, we have included this in Discussion, line 380.
Reviewer 3 Report
Comments and Suggestions for Authors
The article "Oxygen-Enhanced R2* Weighted MRI and Diffusion-Weighted MRI of Head and Neck Squamous Cell Cancer Lymph Nodes in Prediction of 2-Year Outcome Following Chemoradiotherapy" is very interesting. I will make some comments with the intention of improving it.
- Correct the order of references 45 and 46. See line 111 and reference 45, and line 148, reference 46.
- Add references supporting the methodological considerations regarding patient positioning and oxygen delivery (lines 131-134).
- Add current references, including one related to the global burden of disease in cancer.
- Add to the limitations of the study the inherent bias that could be derived from the different treatments received by the patients included in the study.
- Histopathologists and surgeons with expertise have been referred; have they been included in the work team (line 173)?
Thank you
Author Response
1. Correct the order of references 45 and 46. See line 111 and reference 45, and line 148, reference 46.
Response:
Thanks for pointing this out- this has been corrected and references throughout the document updated.
2. Add references supporting the methodological considerations regarding patient positioning and oxygen delivery (lines 131-134).
Response:
This has been added (reference 30).
3. Add current references, including one related to the global burden of disease in cancer.
Response:
We have updated the figures and reference 1 to reflect updated rates etc.
4. Add to the limitations of the study the inherent bias that could be derived from the different treatments received by the patients included in the study.
Response:
We agree- added in discussion lines 381-383
5. Histopathologists and surgeons with expertise have been referred; have they been included in the work team (line 173)?
Response:
Whilst their expertise was relied upon as part of the standard of care multidisciplinary clinical team, they did not directly feed into or review this additional research work.
Round 2
Reviewer 1 Report
Comments and Suggestions for Authors
1. Table 1 is completely inconvenient for perception, patients are not grouped in any way. It is necessary to group patients by localization and present the data in a generalized form. Raw data can be given in additional materials, but not in the text of the manuscript.
2. The authors still have not answered the question about univariate and multivariate risk analysis, which should be added to the manuscript.
3. The manuscript contains a lot of borrowings, the text needs to be reworked.
Author Response
1. Table 1 is completely inconvenient for perception, patients are not grouped in any way. It is necessary to group patients by localization and present the data in a generalized form. Raw data can be given in additional materials, but not in the text of the manuscript.
Response:
We have completely revised the table- it is slightly tricky to include all of the requested information of stage, location and treatment without quite a large table so we have tried to make this as clear as possible and grouped the patients by response and given more information (e.g. p16 status and treatment) for direct overview of the two groups for examination.
2. The authors still have not answered the question about univariate and multivariate risk analysis, which should be added to the manuscript.
Response:
We have undertaken additional logistic regression univariate and multivariate analyses for the largest node per patient data having engaged a statistician to reanalyse this and form models for analysis with odds ratios. This is presented in Table 3; the model was used for the univariate analysis though the multivariate analysis was problematic as described in the text. It was not possible to do this with the all nodes analysis due to multiple sampling per patient, though as described in the text, those data are also analysed by linear mixed method regression therefore the methodology is similar for this analysis.
3. The manuscript contains a lot of borrowings, the text needs to be reworked.
Response:
We have gone through the document to ensure relevant information is referenced and highlighted.
Round 3
Reviewer 1 Report
Comments and Suggestions for Authors
I have no more comments on the manuscript.